# Knockout Mutants of *OsPUB7* Generated Using CRISPR/Cas9 Revealed Abiotic Stress Tolerance in Rice

**DOI:** 10.3390/ijms24065338

**Published:** 2023-03-10

**Authors:** Me-Sun Kim, Seo-Rin Ko, Yu Jin Jung, Kwon-Kyoo Kang, Yung-Jo Lee, Yong-Gu Cho

**Affiliations:** 1Department of Crop Science, College of Agriculture and Life & Environment Sciences, Chungbuk National University, Cheongju 28644, Republic of Korea; 2Division of Horticultural Biotechnology, Hankyong National University, Anseong 17579, Republic of Korea; 3Institute of Korean Prehistory, Cheongju 28763, Republic of Korea

**Keywords:** ubiquitination, E3 ligase, *OsPUB*, CRISPR/Cas9, transcriptomics

## Abstract

Plants produce and accumulate stress-resistant substances when exposed to abiotic stress, which involves a protein conversion mechanism that breaks down stress-damaged proteins and supplies usable amino acids. Eukaryotic protein turnover is mostly driven by the ubiquitination pathway. Among the three enzymes required for protein degradation, E3 ubiquitin ligase plays a pivotal role in most cells, as it determines the specificity of ubiquitination and selects target proteins for degradation. In this study, to investigate the function of *OsPUB7* (Plant U-box gene in *Oryza sativa*), we constructed a CRISPR/Cas9 vector, generated *OsPUB7* gene-edited individuals, and evaluated resistance to abiotic stress using gene-edited lines. A stress-tolerant phenotype was observed as a result of drought and salinity stress treatment in the T_2_ *OsPUB7* gene-edited null lines (PUB7-GE) lacking the T-DNA. In addition, although PUB7-GE did not show any significant change in mRNA expression analysis, it showed lower ion leakage and higher proline content than the wild type (WT). Protein–protein interaction analysis revealed that the expression of the genes (*OsPUB23*, *OsPUB24*, *OsPUB66*, and *OsPUB67*) known to be involved in stress increased in PUB7-GE and this, by forming a 1-node network with *OsPUB66* and *OsPUB7*, acted as a negative regulator of drought and salinity stress. This result provides evidence that *OsPUB7* will be a useful target for both breeding and future research on drought tolerance/abiotic stress in rice.

## 1. Introduction

Intracellular proteins play pivotal roles in signaling, transport, catalysis, membrane fusion, cellular protection, and regulation of biological processes [1]. However, proteins may be damaged in various ways; for example, their tertiary structure may be damaged, they may undergo non-specific aggregation, or there might be imbalance in their homeostasis due to exposure to various stresses [2]. The cyclical process of degrading damaged proteins to generate usable amino acids and resynthesizing them into necessary proteins is called protein turnover, and up to 80% of the proteins in living organisms are degraded by the ubiquitin-proteasome system [3]. The ubiquitin-proteasome system regulates the stability and activity of many proteins and affects various cellular processes in higher plants, including signal transduction, cell division, and responses to biotic and abiotic stresses [4]. It is also a specialized proteolytic system that plays an essential role in controlling protein degradation [5].

Ubiquitin is a small 8.6 kDa protein composed of 76 amino acids that has a C-terminal tail and seven lysine residues. In addition, human and yeast ubiquitins are highly conserved, with 96% sequence identity [6,7]. Ubiquitin plays the most central role in the ubiquitin-proteasome system; it binds to a target protein and acts as a labeling agent. Ubiquitin forms various types of poly-ubiquitin chains, which are detected and degraded by the proteasome [8,9]. The ubiquitin-proteasome system requires a cascade of three major enzymes: ubiquitin-activating enzyme (E1), ubiquitin-conjugating enzyme (E2), and ubiquitin-ligase (E3). The E1 enzyme activates ubiquitin via ATP-dependent reactions, which results in the formation of a high-energy thiol ester bond that connects the glycine at the C-terminal of ubiquitin and the cysteine in E1. The E2 enzyme receives the activated ubiquitin from E1, forms a thiol ester bond, and catalyzes the formation of an iso-peptide bond between ubiquitin and the target protein that is degraded by the E3 enzyme. Finally, the ubiquitinated protein is recognized and degraded by the 26S proteasome [10,11,12].

Compared to the 1 and 15 types of E1 and E2 enzymes known, respectively, hundreds of isoforms of the E3 enzyme have been identified, which detect target proteins for degradation and provide specificity to the ubiquitin-proteasome system [5,13]. E3 ligases are classified into three types: HECT (Homologous to E6-AP Carboxyl Terminus), RING finger (Really Interesting New Gene finger), and U-box, depending on the method via which ubiquitin is linked to the E2 binding domain or the presence of a zinc (Zn) residue [14]. Unlike the RING and U-box types, the HECT type directly attaches ubiquitin to the HECT domain and delivers it to the substrate proteins. The U-box and RING types allow ubiquitin to be directly transferred from E2 to the substrate protein and possess the same folded structure; however, the U-box type does not contain the zinc residue [15,16,17]. U-box proteins are widely present in eukaryotes, and 2, 64, and 77 U-box proteins have been identified in yeast, *Arabidopsis*, and rice, respectively [18,19,20]. The presence of more U-box proteins in highly advanced plants is possibly because the U-box E3 ligase plays an important role in plants; hence, it is subdivided into more types [21]. 

A recent study revealed that plant U-box (PUB) family members are involved in biotic and abiotic stress responses. In *Arabidopsis*, *PUB12* and *PUB13* interact with abscisic acid (ABA)-INTENSITIVE 1 (ABI1), a key PP2C protein involved in ABA signaling, to regulate the response to ABA [22]. The *pub18* and *pub19* double mutants showed resistance to drought stress by enhancing stomatal closure via interaction with *Exo70B1*, which is known to respond to ABA signaling when exposed to ABA [23]. CMPG1-V, a U-box type E3 ligase, contributes to defense responses against powdery mildew and broad-spectrum disease resistance due to an increase in the expression of salicylic acid-responsive genes and accumulation of H_2_O_2_ in plants [24]. The *pub13* mutants have been shown to interact with LYK5 and induce plant immune responses via chitooctaose-induced rapid reactions such as reactive oxygen species production and mitogen-activated protein (MAP) kinase phosphorylation [25]. PUB genes in rice are classified according to the domain structure of the gene, and *OsPUB* genes with U-box/ARM domains are known to be tolerant to abiotic and biotic stress [26]. *OsPUB15* transgenic rice caused severe growth delays and fatal phenotypes in seedling, and transcription levels of *OsPUB15* knockout mutants were shown to increase with H_2_O_2_, salinity, and drought stress [27]. Knockout mutants of *OsPUB16* generated by CRISPR/Cas9 showed improved water resistance by mediating ubiquitination and degradation of *OsMADS23*, a substrate of OSMOTIC STRESS/ABA-ACTIVATED PROTEIN KINASE 9 (SAPK9), and *OsPUB16* over-expression lines showed an increased water deficiency susceptibility to stress [28].

The CRISPR/Cas9 technology is widely used in various organisms because of its ease of use and accuracy. Furthermore, CRISPR/Cas9-mediated target gene editing is used as an important tool for functional analysis of plant and crop genes [29,30,31,32]. 

In this study, we aimed to understand the physiological role of *OsPUB7* with U-box/ARM domain in rice, such as its role in abiotic stress resistance (drought and salinity stress), by generating *OsPUB7* knockout individuals using CRISPR/Cas9 technology. Our study will provide a useful genetic resource for breeding crops that are resistant to abiotic stress caused by climate change.

## 2. Results

### 2.1. Analysis of the Genetic Relationship of OsPUB7 among Plant Species 

Phylogenetic analysis was performed to obtain information regarding the genetic relationship between *OsPUB7* and genes encoding E3 ubiquitin ligases in other plant species. U-box type E3 ubiquitin ligases and other E3 ubiquitin ligases encoded by 26 genes obtained from the NCBI database were analyzed using the MEGA7.0 program (Appendix A). As shown in Figure 1, the E3 ubiquitin ligases were classified according to the conserved protein domains in plant species. Among the genes used in the analysis, one closely related to *OsPUB7* was found to be well conserved in the rice genome. Additionally, PUB genes belonging to wheat (*Triticum aestivum*), barley (*Hordeum vulgare*), and millet (*Setaria italica*) were classified in the same group. These results suggested that rice *OsPUB7*, encoding a domain of the U-box type E3 ubiquitin ligase, exhibits considerably high homology in various plant species, irrespective of plant origin (Figure 1).

### 2.2. Production of OsPUB7-Edited Lines and Analysis of Mutation Types

#### 2.2.1. CRISPR/Cas9-Mediated Editing of *OsPUB7*

To construct gene-edited rice using CRISPR/Cas9 technology, sgRNA was created based on the second exon of *OsPUB7* (Appendix A). Among them, sgRNAs with a GC content of 50%, out of frame score of 84.7, and no mismatches in the genome were selected, which can reduce the off-target probability. The selected sgRNA was located around 113 bp from the 5′-end of the second exon and was analyzed as a target in this region (Figure 2A,B). A Ti-plasmid vector for plant gene editing was constructed using Cas9 under the control of the 35S promoter, sgRNA under the control of the *OsU3* promoter, and *bar* gene as the selection marker. The complete vector, named CRISPR/Cas9-OsPUB7 (Figure 2C), was inoculated into the callus obtained from the seeds of the wild type (WT) plant (*Dongjin*) using an *Agrobacterium*-mediated method, and finally 31 transformants were obtained (Figure 3A). The regenerated plants were transplanted into pots and grown. To confirm the introduction of the T-DNA, polymerase chain reaction (PCR) analysis was performed using the genomic DNA of the T_0_ generation plants (Figure 3B). The results confirmed that all 31 regenerated plants obtained after stable introduction of the CRISPR/Cas9-OsPUB7 vector and transplantation in soil were transgenic.

#### 2.2.2. Analysis of Mutation Types Using Next-Generation Sequencing (NGS)

Genetic mutation types of 31 regenerated individuals were analyzed using NGS technology. Among the 31 transgenic individuals, 15 individuals harbored mutations in *OsPUB7*, and their gene editing efficiency was 48%. The genotypes of the 15 gene-edited individuals were homozygous in 4 individuals, bi-allelic in 6 individuals, and heterozygous in 5 individuals (Appendix A). In homozygous individuals, 1 base was deleted or inserted, and in bi-allelic individuals, 1 base was inserted or 2, 22, or 25 deletions were found. In heterozygous individuals, 1 or 2 bases were inserted (Figure 4; Table 1). Mutations that occurred during double-strand breaking repair in plants appeared mainly as deletions and 1bp insertions. These mutations are stably inherited in offspring in both T_2_ and T_3_ generations according to the classical Mendelian model. When both copies of a target gene are mutated, as in homozygous or bi-allelic editing plants, the genotype is stable and resistant to further editing by CRISPR/Cas9. Bi-allelic gene targeting in seed-propagated plants has the advantage of saving time in correcting genetic alterations [33,34]. T_1_ seeds were harvested by cultivating gene-edited individuals in which the gene mutation type was confirmed. In the T_1_ generation, null individuals lacking the T-DNA were selected using *bar* screening. First, null individuals in which *bar* was completely removed were selected after treatment with 40 ppm basta solution using the *bar* strip test and *bar* PCR (Figure 5). Next, null individuals with a segregation ratio of 3:1 according to the law of segregation (using the chi-square (χ^2^) test) and single copy introduction were confirmed (Table 2). 

### 2.3. Selection of Tolerant Gene-Edited Lines and Examination of Physiological Characteristics According to Abiotic Stress Treatment

#### 2.3.1. Screening of Drought and Salinity Stress

The T_2_
*OsPUB7* gene-edited null lines (PUB7-GE), K343-8-1 and K343-17-8, along with control varieties Dongjin (WT), Sangnambat (T), and Gaya (S), were subjected to drought stress treatment (Figure 6A). As a result of drought stress treatment, all Dongjin, Sangnambat, and Gaya plants stopped growing completely and dried with a damage level of 4 on the 5th day after the treatment. Most plants were permanently dried and did not recover, even after watering. K343-7-5 plants had a similar phenotype to that before stress treatment until the 3rd day of drought stress treatment, and the damage level was 3 to 4 on the 5th day after drought stress treatment. The K343-8-1 and K343-17-8 plants stopped growing and dried; almost all plants showed a damage level of 4 on the 5th day after the drought treatment. However, most of them showed normal growth in the recovery stage and recovered to damage level 2. 

Salinity stress treatment was performed on T_2_ PUB7-GE K343-7-5, K343-8-1, and K343-17-8, as well as on Dongjin (WT), Cheongho (T), and Nampyeong (S) (Figure 6B). As a result of salinity stress treatment, leaf curling and growth retardation were observed in Dongjin, Cheongho, and Nampyeong from the third day after treatment. On the 5th day after treatment, all plants with a damage level of 4 or 3 stopped growing completely and dried. In the recovery stage, the control varieties showed a damage level of 5, and all plants died without recovering. However, in K343-7-5, K343-8-1, and K343-17-8, damage level 2 was observed from the 3rd day after salinity stress treatment, although the damage was lower than that observed in Dongjin and Cheongho. Contrary to that observed in the control varieties in the recovery stage, K343-8-1 and K343-17-8 showed almost normal growth, with a damage level of 2; K343-7-5 recovered to damage level 3. Overall, the *OsPUB7* knockout mutants exhibited an increase in the abiotic stress tolerance of rice. 

#### 2.3.2. Determination of Physiological Characteristics and Analysis of mRNA Expression

To confirm the degree of tolerance to stress treatment in terms of physiological traits, T_2_ PUB7-GE were subjected to ion leakage and proline content analyses. The degree of ion leakage before and after drought stress treatment was analyzed in three replicates using 100 mg leaves. Results showed that ion leakage with cell damage had the following order: Gaya (S, 62.37%), Dongjin (WT, 44.19%), Sangnambat (T, 42.19%), K343-8-1 (24.03%), and K343-17-8 (12.86%) on the 5th day after drought stress treatment. In addition, T_2_ PUB7-GE maintained a low level of ion leakage during recovery, with a tendency similar to that observed before recovery (Figure 7A). To confirm the degree of tolerance to drought stress treatment, proline content analysis was performed on the 3rd day, when the phenotype started to appear (Figure 7B). The results showed that the proline content of WT was 1.001 mg/g, whereas that of K343-7-5 was 1.108 mg/g, that of K343-8-1 was 2.160 mg/g, and that of K343-17-8 was 2.011 mg/g, which were approximately 10–110% more than that of the WT. 

The ion leakage analysis of T_2_ PUB7-GE and that of the salinity stress control varieties, Cheongho (T) and Nampyeong (S), showed that ion leakage followed the order of Dongjin (WT, 64.03%), Nampyeong (S, 55.51%), Cheongho (T, 33.33%), K343-8-1 (24.33%), K343-7-5 (23.19%), and K343-17-8 (11.71%) on the 5th day after salinity stress treatment (Figure 7C). To observe the degree of tolerance to salinity stress treatment, proline content was analyzed on the 3rd day, when the stress phenotype started to appear (Figure 7D). The proline content of WT was 0.789 mg/g, whereas that of K343-7-5 was 1.093 mg/g, that of K343-8-1 was 0.963 mg/g, and that of K343-17-8 was 0.994 mg/g, which were 22–38% more than that of Dongjin. Overall, considering the results of ion leakage and proline content analyses, we concluded that T_2_ PUB7-GE showed tolerance to drought and salinity stress.

To survey the expression level of *OsPUB7* during abiotic stress treatment, qRT-PCR analysis was performed using the leaves 0, 3, 6, 12, 24, and 36 h after drought and salinity stress treatment (Figure 8A). At 3 h after drought stress treatment, T_2_ PUB7-GE appeared to show the highest gene expression, although finally, the expression did not differ significantly between WT and PUB7-GE or between individuals. This suggested that although *OsPUB7* harbored a mutation introduced using the CRISPR/Cas9 technology, the transcriptional regulatory region of the gene functioned normally. Transcription of *OsPUB7* occurs normally, although translation is believed to be restricted due to the creation of an early stop codon (due to a codon frame shift caused by an InDel). Similar to that observed for drought stress treatment, RNA expression did not differ significantly between WT and PUB7-GE, nor was any specific trend in expression change observed after salinity stress treatment (Figure 8B).

### 2.4. Analysis of Protein Structure in PUB7-GE 

Before confirming the codon frame shift in T_2_ PUB7-GE, the genotypes of the lines showing tolerance to abiotic stress were analyzed. Results confirmed that all bi-allelic genotypes in the T_0_ generation were fixed as homozygous genotypes in the T_2_ generation, and the mutation types were identified as 2-bp deletion and 1-bp insertion, respectively (Figure 9). 

Analysis of the amino acid sequence of three T_2_ PUB7-GE, which showed tolerance to drought and salinity stress, revealed that an early stop codon was generated at 159 bp (Figure 10B,D) and 162 bp (Figure 10C) from the 5′ end of the second exon, due to a codon frame shift. We assumed that the function of *OsPUB7* was lost because translation was halted by the generation of the early stop codon. The protein structure of T_2_ PUB7-GE with amino acid mutations was analyzed, and it was observed that, compared to the WT, 308 amino acids (Figure 11B,D) and 307 amino acids (Figure 11C) of *OsPUB7* were not translated. Hence, we believe that *OsPUB7-*edited null lines lost the E3 ubiquitin ligase activity, as normal protein synthesis did not occur (Figure 11). 

### 2.5. Analysis of Protein–Protein Interactions

We aimed to identify the interaction between genes related to abiotic stress tolerance [35] and *OsPUB7*. Comparison of gene expression levels using T_2_ PUB7-GE mRNA derived after stress treatment revealed that the mRNA levels of the four genes, *OsPUB23*, *OsPUB24*, *OsPUB66*, and *OsPUB67*, increased (Figure 12). String DB-based protein interaction analysis between *OsPUB7* and *OsPUB* families revealed that a 1-node network was formed with *Os*PUB66 (Figure 13). This indicated the functional association between *OsPUB* genes; it can be inferred that *OsPUB7* acts as a negative regulator in WT and that *OsPUB23*, *OsPUB24*, *OsPUB66*, and *Os*PUB67 act to show tolerance to drought and salinity stress after knocking out *OsPUB7*.

### 2.6. Analysis of RNA-Sequencing Data of PUB7-GE

To identify the genes related to the abiotic stress tolerance pathway in PUB7-GE, we performed RNA-sequencing analysis in PUB7-GE K343-8-1 and compared it to that of the WT plants (Appendix A). We estimated the expression levels of transcripts using fragments per kilobase of exon per million mapped fragment (FPKM) values [36] (Appendix A). A gene was considered expressed in a sample if its FPKM was greater than 1. In total, 44,759 genes were expressed in at least one of four samples analyzed. Among the genes whose expression was confirmed, 27,831 genes were constitutively expressed in all samples, with a coefficient of variation <10%. Four samples were classified as harboring highly expressed genes under drought and salinity stress conditions. Differentially expressed gene (DEG) analysis revealed that the *OsPUB7* knockout line contained 15,433 up-regulated genes and 17,162 down-regulated genes under drought stress treatment compared to that in the WT. Furthermore, K343-8-1 contained 15,654 up-regulated genes and 17,304 down-regulated genes under salinity stress treatment (Figure 14A). All of the differentially expressed genes were categorized into three main functional subgroups (Appendix A), including Molecular Function (4789), Biological Process (4404), and Cellular Components (4401). For the Cellular Components category, the nucleus (GO:0005634) and integral component of membrane (GO:0016021) were the most highly represented categories. For the Molecular Function category, genes associated with metal ion binding (GO:0046872) and ATP binding (GO:0005524) have the highest fractions. Under the Biological Process category, proteins related to regulation of transcription (GO:0006355) and defense response (GO:0006952) were most frequent. After analyzing the down-regulated genes to determine the expression pattern of interacting genes in *OsPUB7* and *OsPUB* gene families (Appendix A), 5 out of 62 *OsPUB* genes were found to be down-regulated after abiotic stress treatment (Figure 14B). In addition, 6 of the 63 *OsPUB* genes were found to be up-regulated in *OsPUB7-*edited plants (Figure 14C). The up-regulation of *OsPUB* genes affected by the *OsPUB7* knockout under drought and salinity stress treatment conditions suggests that *OsPUB* genes are closely related to abiotic stress responses. Based on these criteria, we finally identified six genes as the target candidates for the *OsPUB7*-related abiotic stress tolerance pathway.

To validate the results of sequencing analysis, qRT-PCR was performed using six *OsPUB* genes, which are abiotic stress-responsive genes selected from the in silico analysis of DEGs (Figure 15). Interestingly, RNA-sequencing analysis confirmed that, with the exception of *OsPUB12*, the remaining five genes showed significantly higher expression levels. The results of qRT-PCR of all six genes showed an up-regulated expression pattern in the *OsPUB7*-edited line. 

### 2.7. Investigation of Agronomic Traits 

To observe the growth of T_3_ PUB7-GE in a rice field, major agricultural traits such as plant height, culm length, panicle length, and number of tillers were investigated (Table 3, Figure 16). The plant height and number of tillers of K343-7-5 did not differ to those of WT, although culm length was approximately 4.8 cm shorter and panicle length was approximately 5 cm longer. K343-8-1 did not differ from the WT in terms of the four agricultural traits. The culm length, panicle length, and number of tillers of K343-17-8 did not differ from those of the WT, although plant height was approximately 4.8 cm longer than the WT. Despite some differences between *OsPUB7*-edited lines, overall, the agricultural traits were similar to those of WT, suggesting that gene editing using the CRISPR/Cas9 technology did not significantly affect genes other than *OsPUB7*.

## 3. Discussion

Target gene editing using the CRISPR/Cas9 technology has been applied to many crops for studying gene function and for improving agricultural traits such as resistance to various stress factors. CRISPR/Cas9-mediated gene editing is simple and easy to use [37], and consumers’ concerns regarding genetically modified organisms can also be avoided by removing foreign gene regions (T-DNA) introduced via selfing [38]. Hence, in this study, we have studied the physiological role of *OsPUB7* in response to drought and salinity stress using CRISPR/Cas9-based genome-edited rice plants (PUB7-GE).

Drought and salinity stress treatment of these null lines revealed that in T_2_ PUB7-GE (K343-7-5 and K343-8-1), K343-17-8 showed tolerance to drought and salinity stress. In addition, analysis of the degree of abiotic stress and physiological tolerance performed by measuring ion leakage and proline content revealed that the tolerance pattern was the same as the observed phenotype. We observed that *OsPUB7* acted as a negative regulator of drought and salinity stress and that the *OsPUB7* knockout enhanced abiotic stress tolerance in rice. In the CRISPR/Cas9 system, translation of the target gene is halted when a premature termination codon (PTC) exists in the coding region of the gene to be corrected [39]. Sequencing analysis results showed that the mutation types in PUB7-GE used for drought and salinity stress treatment included a 2-bp deletion (K343-2-5 and K343-17-8) and 1-bp insertion (K343-8-1), respectively. Owing to a frame shift caused by an InDel in the coding region of *OsPUB7*, a PTC was formed at the beginning of the second exon sequence, as a result of which the PUB7-GE lines, K343-7-5, K343-18-8 and K343-8-1 were able to synthesize only 151 and 152 amino acids, respectively. PCT can occur during translation in the cytoplasm due to a codon frame shift caused by InDel in a gene. During this, when an mRNA binds to the ribosome and the stop codon is recognized, the ribosome degrades the mRNA via non-sense-mediated mRNA decay (NMD) to eliminate the potential risk associated with the accumulation of truncated proteins [40,41]. In the case of *OsPUB7* gene editing lines, the frame shift-induced PCT is present at the beginning of the nucleotide sequence of the second exon of *OsPUB7*, which not only prevents subsequent amino acid synthesis, but also prevents production of the truncated protein. Protein–protein interaction analysis was performed with *Os*PUB23, *Os*PUB24, *Os*PUB66, *Os*PUB67, and *Os*PUB7, which are known to be involved in drought tolerance based on the results of transcriptome analysis under drought and salinity stress conditions. We confirmed that the expression of four genes increased in T_2_ PUB7-GE compared to that in the WT. The functions of Group II members in the PUB family of *Arabidopsis* have been extensively investigated in plant abiotic stress treatment. *AtPUB22/AtPUB23* are negative regulators that mediate drought response in an ABA-independent pathway [42]. It has been reported that AtPUB30 negatively regulates salinity tolerance by promoting *BRI1 KINASE INHIBITOR 1 (BKI1)* degradation [43]. It can be assumed that the *OsPUB7* gene in WT can act as a negative regulator of these four genes, and knockout of *OsPUB7* affects the expression of several *OsPUB* genes, enhancing the drought and salinity tolerance of rice. To identify genes related to the abiotic stress tolerance pathway in the *OsPUB7* knockout line, we performed RNA-sequencing analysis in PUB7-GE K343-8-1 and compared the results with those obtained using the WT plants. The up-regulated genes were analyzed to determine the expression pattern of interacting genes between the *OsPUB7* and the *OsPUB* gene family. Results revealed that 6 out of 63 *OsPUB* genes were up-regulated after abiotic stress treatment. To validate the results of sequencing analysis, qRT-PCR was performed using six *OsPUB* genes, which are abiotic stress-responsive genes selected from the in silico analysis of DEGs. In qRT-PCR, five out of the six genes that showed up-regulated expression pattern in the *OsPUB*-edited line were found to be significantly highly expressed. This study provided clues regarding the interaction and linking role between *OsPUB7* and a small number of *OsPUB* genes (*OsPUB66*, *OsPUB23*, *OsPUB24*, *OsPUB67*) under drought and salinity stress treatment. Among the up-regulated genes, two OsARM genes (*OsPUB22* and *OsPUB24*) were differentially expressed in drought and salinity stress treatments. The ARM repeat (Armadillo repeat) gene has an ARM repeat domain consisting of one short α-helix and two long α-helices [20,44]. ARM genes have diverse functions such as protein degradation and signal transduction, nuclear transport, and cell adhesion [45]. In addition, it was confirmed that among up-regulated genes, genes encoding two protein kinases (*OsPUB66* and *OsPUB67*) were over-represented. Most of these kinases are receptors such as Pelle kinase, and these receptors are the largest gene families in *Arabidopsis* and rice and respond to various abiotic and biotic stresses [46]. In rice, the receptor-like cytoplasmic kinase *GROWTH UNDER DROUGHT KINASE (GUDK)* has been shown to improve resistance to drought stress through activation of transcription factor *APETALA2/ETHYLENE RESPONSE FACTOR OsAP37* [47]. In *Arabidopsis*, it was confirmed that water use efficiency is improved through the overexpression of *Leucine-Rich Repeats Receptor-Like Kinase (LRR-RLK)* genes [48]. Regulation of ubiquitination levels and stability in the changing expression of *OsPUB7*, and the signaling actions of sub-materials such as abscisic acid (ABA) and jasmonic acid (JA), should be explored in future work.

In conclusion, the editing of *OsPUB7* using CRISPR/Cas9 technology resulted in the identification of several molecular mechanisms related to stress response that involve *OsPUB* gene-related pathways. These genes will be useful targets for both breeding and future research on drought tolerance/abiotic stress in rice. In addition, *OsPUB7* can be expected to interact with the genes related to both biotic and abiotic stress, which may result in the degradation of damaged proteins to provide free amino acids for the synthesis of other proteins.

## 4. Materials and Methods

### 4.1. Plant Materials

In this study, Dongjinbyeo, a rice variety that is not tolerant to drought and salinity stress, was used as plant material for the generation of transgenic rice using CRISPR/Cas9-OsPUB7 vector. Seedlings were transplanted in a glass greenhouse and experimental farm at 30 × 15 cm spacing, with one seedling per hill arranged in an incomplete block design. The fertilizer N-P_2_O_5_-K_2_O were applied at a rate of 90-45-47 kg/ha. Cultivation management was performed following the rice cultivation standards adapted to the experimental area of Chungbuk National University.

### 4.2. Analysis of Phylogenetic Tree

To infer the relationship between *PUB7* in rice and Arabidopsis and other crops, we calculated evolutionary divergence estimates and constructed a phylogenetic tree. The *OsPUB7* gene was queried in RAP-DB(https://rapdb.dna.affrc.go.jp/index.html accessed on 3 August 2020), and the amino acid sequences of other plant gene homologues related to OsPUB7 were collected from NCBI (https://blast.ncbi.nlm.nih.gov accessed on 3 August 2020) using the OsPUB7 protein sequence. Corresponding genomic sequences and coding region sequences were blast searched in Gramene (http://www.gramene.org accessed on 3 August 2020), Rice Genome Annotation Project (rice.plantbiolog.msu.edu accessed on 3 August 2020), and NCBI (http://www.ncbi.nlm.nih.gov/gene accessed on 3 August 2020). The evolutionary history was inferred using the neighbor-joining method [49], and the phylogenetic tree was constructed and viewed with MEGA 7 software [50].

### 4.3. Selection of Target Sequences and Vector Construction

Target sites and sgRNAs for the 1000 bp exon region of OsPUB7 (Os04g0348400) adjacent to a protospacer-adjacent motif (PAM) were designed using the CRISPR RGEN tool (http://www.rgenome.net/ accessed on 2 September 2020) developed by the Hanyang University ([51]. According to the principles for designing target sequences in the CRISPR/Cas9 system, 20 base pairs excluding the PAM sequence of the selected sgRNA were designed as candidate target sequences as follows: target-up, 5′-ggcaGTGTGCACCACTGTGAGATCATGG-3′; target-down, 5′-aaacCCATGATCTCACAGTGGTGCACAC-3′. A vector for rice transformation was constructed to introduce the created OsPUB7 RGEN into the WT line (*Oryza sativa* L. cv. Dongjin-byeo). The pPZP-3′PINII-Bar vector was used as the parent vector for plant transformation. Cas9, under the CaMV 35S promoter, and sgRNA, under the OsU3 promoter, were ligated to this vector. Bar, under the control of the CaMV 35S promoter, was used as the selection marker; it was first linked to p35S at the HindIII/SacI site of the pPZP-3′PINII-Bar vector, which was then digested with BamHI/SacI and ligated with Cas9. Finally, AarI and XhoI sites were added to connect the sgRNA to the OsU3 promoter (Figure 2). The vector constructed was named CRISPR/Cas9-OsPUB7. The binary constructs were then introduced into Agrobacterium tumefaciens strain EHA105 via electroporation [52].

### 4.4. Agrobacterium-Mediated Transformation and Analysis of Mutation Types Using NGS

*Agrobacterium*-mediated transformation of embryogenic callus was performed by Lee et al. [52]. After 4 weeks of rooting, the regenerated rice plants were transferred to pots in a greenhouse maintained at 30 °C during the day and at 23 °C at night. Genomic DNA was extracted from the leaves of T_0_ individuals confirmed to harbor the T-DNA for gene mutation type analysis and mutant selection. Deep-sequencing was performed to amplify the genomic region containing the CRISPR/Cas9 target sites using specific primers adjacent to the designed target site [53]. PCR amplicons were sequenced by forming paired-end reads using MiniSeq (Illumina, San Diego, CA, USA). The NGS data obtained were analyzed using Cas-Analyzer (https://www.rgenome.net/cas-analyzer accessed on 5 April 2021) [54]. Reads that occurred only once were excluded to remove errors associated with amplification and sequencing. Insertion and deletion mutations were considered as mutations induced by Cas9. To select null lines without T-DNA, 40 ppm of basta solution was used to treat individuals that generated more than six leaves after the T_1_ generation. The basta solution was embedded in a gauze and placed on a part of the seedling leaves grown to the six-leaf stage, which were observed after 7 days to distinguish between resistance and sensitivity. These plants were also used to reconfirm whether they were resistant or sensitive using the bar strip test. T-DNA-free plants were selected on the basis of the death of the leaves after basta treatment and the absence of bands in the *bar* strip experiments. Finally, PCR analysis was performed on null lines clearly lacking the T-DNA using a *bar* primer, and individuals with no band in 1% agarose gel electrophoresis were selected as the final null T_1_ lines.

### 4.5. Screening for Drought and Salt Stress Resistance

For screening drought stress tolerance, 4-week-old T_2_ PUB7-GE lines, along with drought-tolerant Sangnambat and susceptible Gaya, were grown in the greenhouse and transplanted to soil with 5% moisture content. Then the screening was performed without watering until the appearance of the drought phenotype. The soil moisture content was measured using a soil moisture meter (ProCheck, Armidale, NSW). The screening was stopped when the susceptible Gaya line showed clear signs of damage, following which it was irrigated to recover for 2 weeks.

For the screening of salinity stress tolerance of the GE null lines, 4-week-old plants of salt-tolerant Cheongho and susceptible Nampyeong were treated with 250 mM sodium chloride (NaCl). NaCl solution was replaced once every two days to prevent the reduction of salt concentration due to absorption and evaporation during treatment. The screening was stopped when the susceptible Nampyeong showed clear signs of damage. The plant was then irrigated with clean water and allowed to recover for 2 weeks. Leaf rolling and leaf drying were evaluated [55] when the susceptible variety showed symptoms under stress. The seedlings were evaluated using the scales described in IRRI [56]: 1 = normal growth and no leaf symptoms, 2 = nearly normal growth although tips of few leaves are discolored and rolled, 3 = growth is severely retarded with most leaves rolled and only a few elongating, 4 = complete cessation of growth along with drying of some leaves and death of some plants, and 5 = almost all plants died (Figure 17).

### 4.6. Measurement of the Physiological Parameters of Transgenic Rice

The proline content in leaves was estimated according to the method used by Abdula et al. [57], with some modifications. Briefly, 0.1 g of rice leaves was ground with 1 mL MCW buffer (MeOH:chloroform:water = 12:5:1), and the homogenate was centrifuged at 4 °C and 13,000 rpm for 10 min. The extract was mixed with 200 μL of 2% ninhydrin reagent and 200 μL of acetic acid. The mixture was mixed using a vortex mixer and boiled at 100 °C for 45 min. The boiled sample was then frozen on ice, combined with 1000 μL toluene, and then left to stand for 5–10 min. Absorbance of the reddish pink upper phase was recorded at 520 nm against a toluene blank. L-proline was used as the standard solution.

Ion leakage was measured before and after autoclaving using 0.1 g leaves (1 cm in diameter). The leaf discs were placed in closed tubes containing 10 mL distilled water and incubated at 32 °C for 2 h. Subsequently, the initial electrical conductivity of the solution (EC1) was determined using a multi-range EC meter (HANNA Instruments, Woonsocket, RI, USA). The samples were autoclaved to release all electrolytes and cooled down to 25 °C, and their final electrical conductivity (EC2) was measured. The electrolyte leakage (EL) was calculated as EL = (EC1/EC2) × 100 (%).

### 4.7. mRNA Expression Analysis

Leaf samples were collected at 0, 3, 6, 12, and 24 h after treatment and at 72 h after recovery. Total RNA was extracted from leaf tissues using the RNeasy plant mini kit (QIAGEN, Redwood City, USA) according to the manufacturer’s instructions. The relative purity and concentration of RNA were estimated using NanoDrop One (Thermo Fisher Scientific, Wilmington, USA) and stored in a −80 °C freezer. The first-strand cDNAs were synthesized using Oligo (dT)20 primer and ReverTra Ace^TM^ qPCR RT master mix (TOYOBO, Osaka, Japan). Actin primers were used as the internal control to normalize the results of the real-time qRT-PCR. All samples were analyzed in triplicate to increase the accuracy of the experiment.

### 4.8. Analysis of OsPUB7-GE

Codon changes caused by frame shift mutations were confirmed from the nucleotide and amino acid sequence analysis of PUB7-GE-harboring InDels. For the amino acid sequence, the translator tool (https://web.expasy.org/translate/ accessed on 5 December 2021) provided by Expasy was used. Based on the amino acid sequence of PUB7-GE, the protein structure was confirmed using SBI’s modeling program (https://swissmodel.expasy.org/interactive accessed on 5 December 2021).

### 4.9. Protein–Protein Interaction Network Analysis

Protein–protein interactions were predicted using the STRING database (https://string-db.org accessed on 5 August 2021) to investigate interaction between the U-box type E3 ubiquitin ligase genes and *OsPUB7* in rice. The OsPUB7 amino acid sequence obtained from RAP-DB (https://rapdb.dna.affrc.go.jp accessed on 5 August 2021) was used in the amino acid sequence column, and orthologs of *Oryza sativa* were selected as references. After completing the BLAST step, the network was constructed using the highest score gene.

### 4.10. RNA-Sequencing Analysis

Leaves of PUB7-GE and WT lines were collected after drought stress treatment. RNA libraries were sequenced using the Illumina High-Seq 2500 platform, provided by a commercial service provider (Theragen Bio, Seongnam, Korea). The low-quality bases (Q < 15) were trimmed at both ends of the sequence using a customized program, and the adapter was trimmed using Cutadapt. The high-quality reads were subsequently aligned to the IRGSP-1.0 reference genome sequence using Bowtie and TopHat. The expression levels of each transcript were expressed as FPKM values calculated based on the number of mapped reads. All DEGs were determined using Cufflinks program v.2.0.1 (http://cufflinks.cbcb.umd.edu accessed on 1 January 2022) [58]. All DEGs were identified using DEGseq and categorized according to the Gene Ontology framework using DAVID bioinformatics resources (ver. 6.8) (https://david,ncifcrf.gov/tools.jsp accessed on 1 January 2022).

### 4.11. Statistical Analysis

Data were analyzed by one-way analysis of variance (ANOVA) using Statistical Analysis System (SAS version 9.4). Values are mean ± SE (*n* = 3) and statistical significance was set to *p* < 0.05 according to Duncan’s multiple range test.

## Figures and Tables

**Figure 1 ijms-24-05338-f001:**
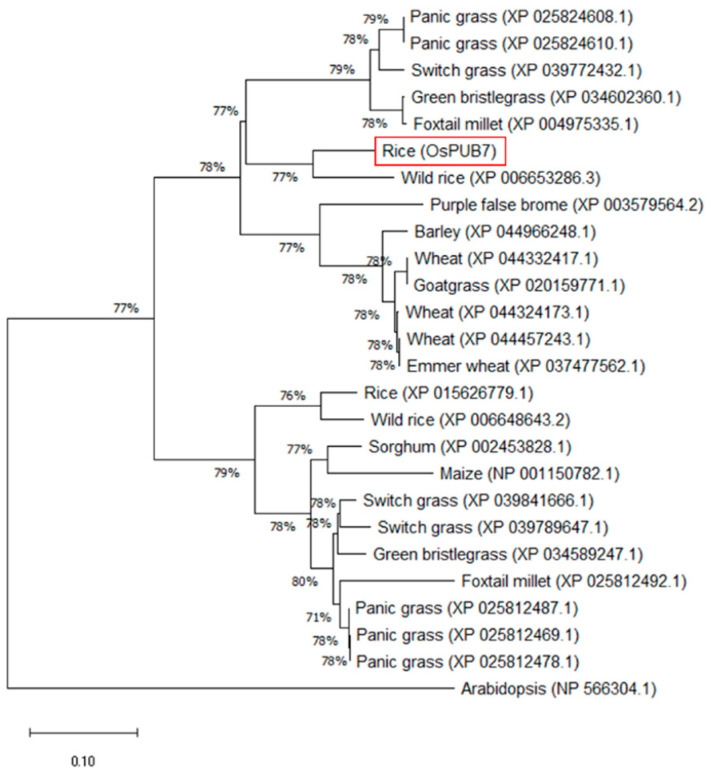
Phylogenetic analysis of *OsPUB7* in various plant species. The phylogenetic tree was constructed using the neighbor-joining method of MEGAX. The numbers represent bootstrap values from 1000 replicates.

**Figure 2 ijms-24-05338-f002:**
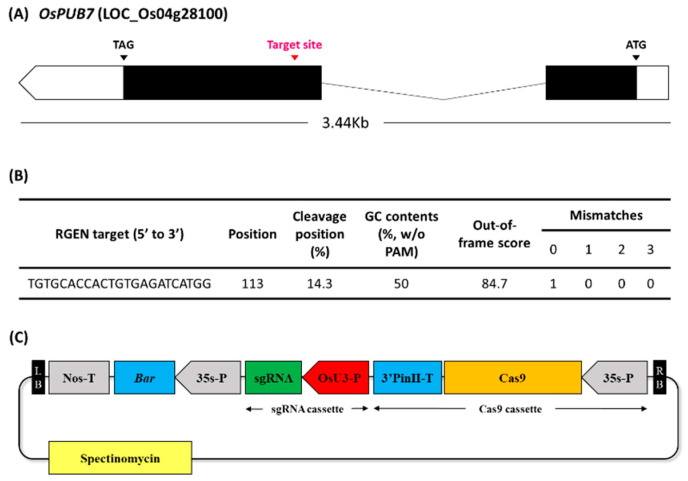
Designing of sgRNA based on CRISPR RGEN Tool (http://www.rgenome.net/ accessed on 2 September 2020). (**A**) Selection of the target region of *OsPUB7* using the CRISPR RGEN tool. (**B**) The position of the sgRNA target sites in the *OsPUB7* sequence. (**C**) Construction of the Ti-plasmid vector of the sgRNA region for CRISPR/Cas9-mediated mutagenesis of *OsPUB7* in rice. LB, left border; RB, right border; Nos, nopaline synthase terminator; Bar, basta resistance gene; sgRNA, single guide RNA; *OsU3*-P, *Oryza sativa* U3 promoter driven sgRNA cassette; Cas9, human codon-optimized Cas9 expressing cassette.

**Figure 3 ijms-24-05338-f003:**
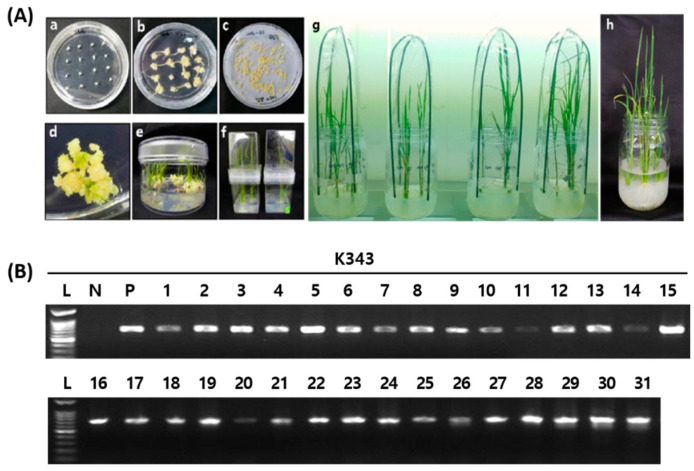
Development of transgenic rice plants with CRISPR/Cas9-OsPUB7 vector using *Agrobacterium*-mediated transformation. (**A**) *Dongjin* (WT) transformed with pPZP-3′PinII-Bar vector containing the CRISPR/Cas9:sgRNA gene: (**a**) seeds plated on 2N6 media; (**b**) callus formation; (**c**) *Agrobacterium* co-culture in 2N6 AS media; (**d**,**e**) multi-shoot differentiation; (**f**) regenerated plants in rooting medium; (**g**,**h**) acclimation in tissue culture room and planted in soil. (**B**) Detection of sgRNA/Cas9-mediated DNA modifications using PCR analysis of *bar* and *nos* terminator regions. L, DNA ladder; N, negative control (sterile water); P, positive control (plasmid vector).

**Figure 4 ijms-24-05338-f004:**
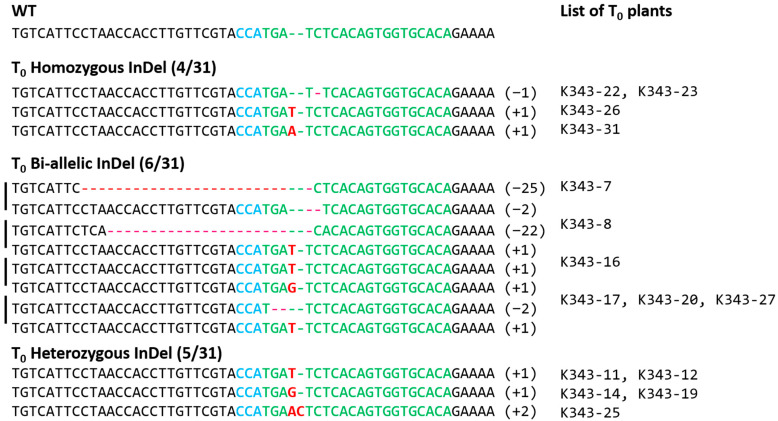
Identification of mutation genotypes in *OsPUB7* generated using CRISPR/Cas9 technology using NGS analysis in *Dongjin* (WT) and T_0_ plants. Green indicates the target sequence, and blue indicates the PAM region. The InDel sequence of the target locus is shown in red. Deletion in the analysis region is indicated by “-”, and insertion is indicated by an insertion of nucleotide such as “T” or “G”. The minus (−) and plus (+) signs indicate the number of nucleotides deleted and inserted at the target sites, respectively. WT, wild type sequence with no mutations.

**Figure 5 ijms-24-05338-f005:**
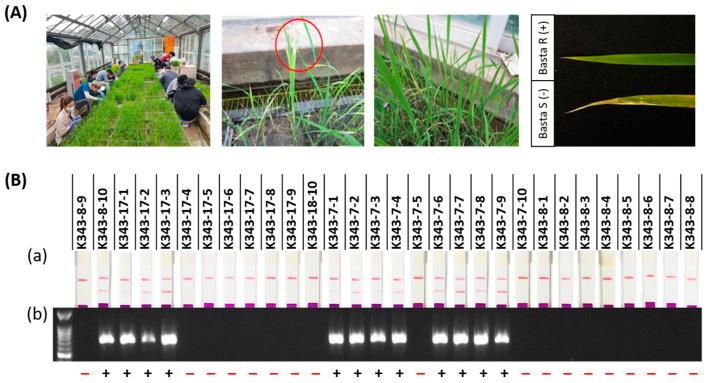
Selection of T-DNA-free null lines. *Bar* screening after glufosinate/basta treatment on leaf tip, and *bar* strip analysis in T_1_ generation. (**A**) Screening of *OsPUB7* transgenic T_0_ mutants after treatment with 40 ppm basta. (**B**) Detection of the selectable marker gene using *PAT/bar* test strip and *bar* PCR: (**a**) selection of *OsPUB7* null lines after *PAT/bar* strip test; (**b**) selection of *OsPUB7* transgenic T_1_ plants after *bar* PCR. The minus (−) and plus (+) signs indicate resistance to *bar*.

**Figure 6 ijms-24-05338-f006:**
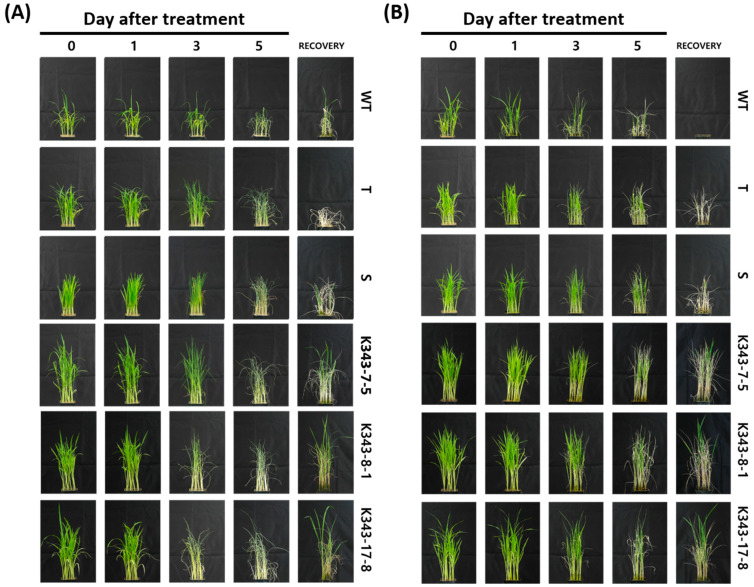
Phenotype of responses induced by drought and salinity stress treatment in T_2_ PUB7-GE. (**A**) Phenotype of PUB7-GE in the presence of 5% soil moisture content. WT, Dongjin; T, Sangnambat; S, Gaya. (**B**) Phenotype of PUB7-GE in the presence of 250 mM NaCl. WT, Dongjin; T, Cheongho; S, Nampyeong.

**Figure 7 ijms-24-05338-f007:**
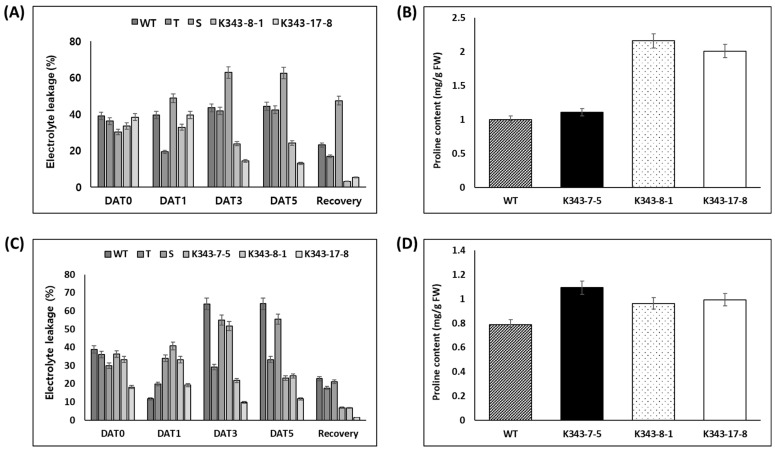
Determination of physiological traits in T_2_ PUB7-GE during drought and salinity stress treatment. (**A**) Ion leakage levels (%) were measured after 5 days of drought stress treatment and after 3 weeks of recovery in WT and PUB7-GE. (**B**) Proline contents (mg/g FW) of WT and PUB7-GE after drought treatment (**C**) Ion leakage levels (%) were measured after 5 days of salinity stress treatment and after 2 weeks of recovery in WT and PUB7-GE. (**D**) Proline contents (mg/g) of WT and PUB7-GE after salinity treatment. Three replicates were used, and data are shown with SE.

**Figure 8 ijms-24-05338-f008:**
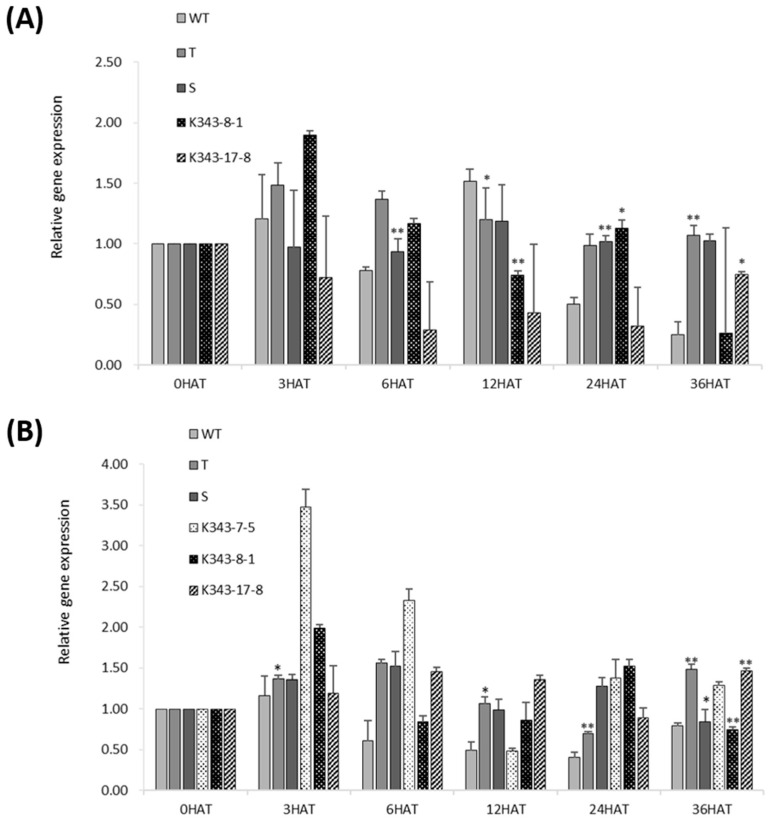
RNA expression patterns of *OsPUB7* in response to drought and salinity stress treatment analyzed using qRT-PCR. Relative gene expression level was normalized to that of the rice actin gene, *ACT1*. Values are presented as mean ± standard error. Three independent biological replicates were used. Bars represent standard deviations (SD) of three technical replicates. Significant differences according to *t*-test are indicated using asterisks (*, *p* < 0.05; **, *p* < 0.01).

**Figure 9 ijms-24-05338-f009:**
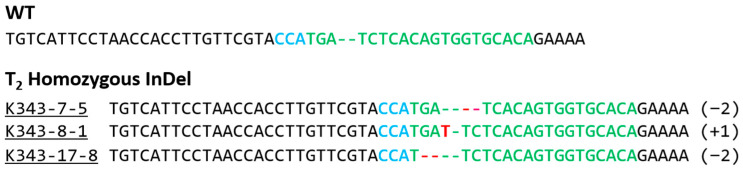
Identification of genotypes using NGS analysis in T_2_ PUB7-GE. The minus (−) and plus (+) signs indicate the number of nucleotides deleted and inserted at the target sites, respectively. WT, wild type sequence with no mutations.

**Figure 10 ijms-24-05338-f010:**
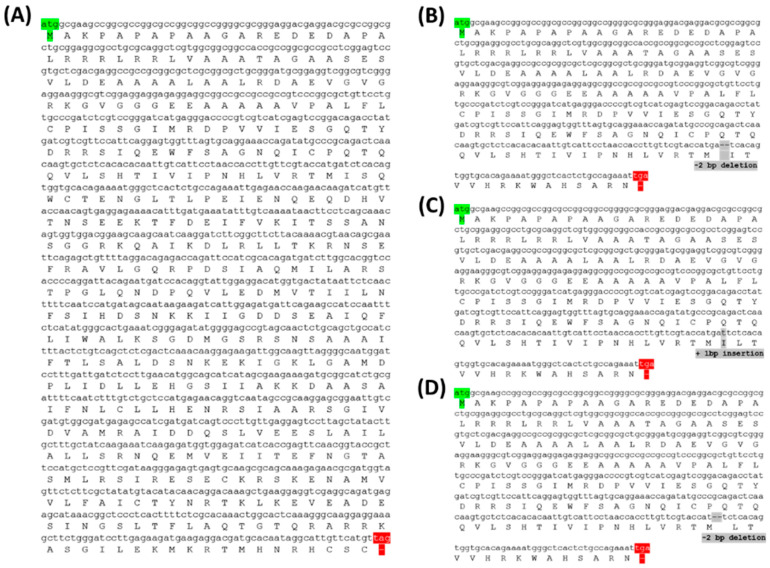
Amino acid sequence based on nucleotide sequence of PUB7-GE. Green indicates the start codon, red indicates the stop codon, and gray indicates the InDel mutation type. (**A**) *OsPUB7* wild type. (**B**) PUB7-GE K343-7-5. (**C**) PUB7-GE K343-8-1. (**D**) PUB7-GE K343-17-8.

**Figure 11 ijms-24-05338-f011:**
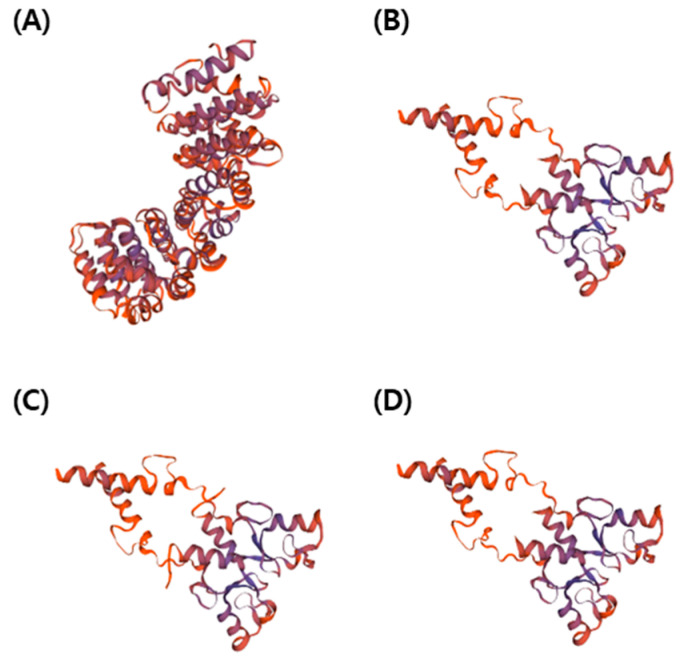
3D protein structure based on amino acid sequences of the gene-edited PUB7-GE. (**A**) *OsPUB7* wild type. (**B**) PUB7-GE K343-7-5. (**C**) PUB7-GE K343-8-1. (**D**) PUB7-GE K343-17-8.

**Figure 12 ijms-24-05338-f012:**
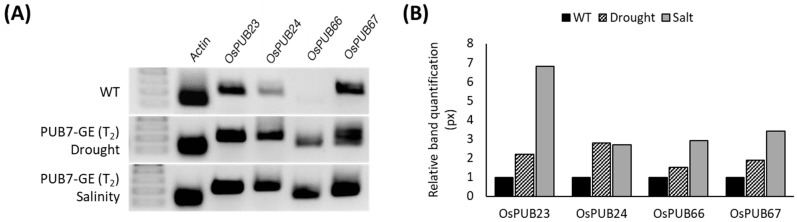
Expression analysis of *OsPUB23*, *OsPUB24*, *OsPUB66*, and *OsPUB67* after drought and salinity stress treatment in T_2_ PUB7-GE. (**A**) Confirmation of the mRNA expression of stress-related genes in PUB7-GE 12 h after stress treatment using agarose gel electrophoresis. (**B**) Band intensity was quantified using ImageJ.

**Figure 13 ijms-24-05338-f013:**
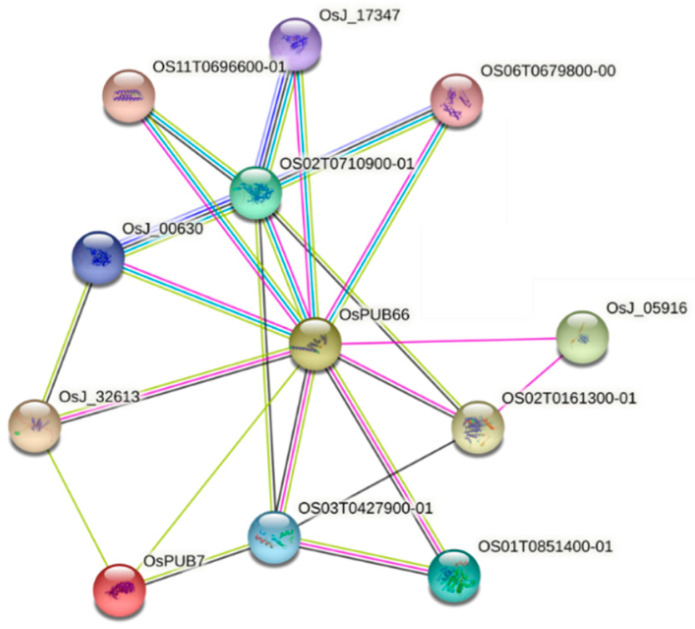
Result of protein–protein interaction analysis obtained using String DB analysis.

**Figure 14 ijms-24-05338-f014:**
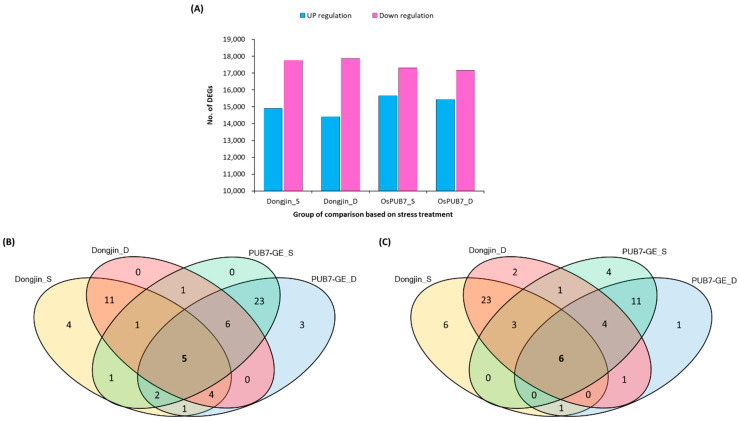
Comparative analysis of differentially expressed genes (DEGs) after drought and salinity stress treatment. (**A**) Number of down-regulated and up-regulated DEGs in samples in response to stress treatment. (**B**) Venn diagram of down-regulated genes in PUB7-GE (K343-8-1) vs. WT (*Dongjin*). (**C**) Venn diagram of up-regulated genes in PUB7-GE (K343-8-1) vs. WT (*Dongjin*).

**Figure 15 ijms-24-05338-f015:**
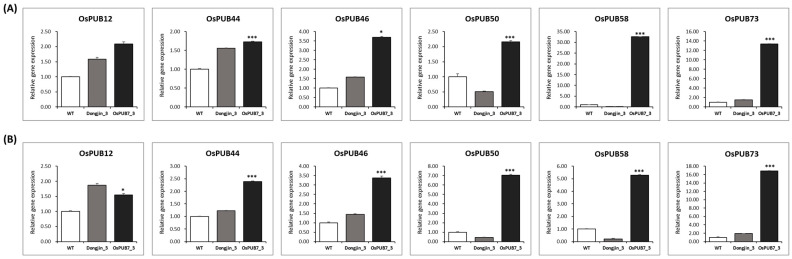
Relative gene expression determined using qRT-PCR. (**A**) Gene expression of six candidate differentially expressed genes (DEGs) in WT and PUB7-GE (K343-8-1) after drought stress treatment. (**B**) Expression of six candidate DEGs in WT and PUB7-GE (K343-8-1) after salinity stress treatment. Error bars indicate SEs (*n* = 3, three biological replicates). Values for WT and PUB7-GE differed significantly. *, *p* < 0.05; ***, *p* < 0.001.

**Figure 16 ijms-24-05338-f016:**
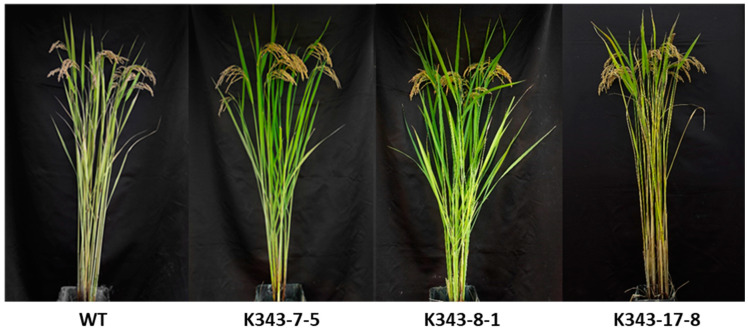
Phenotypes of selected PUB7-GE in the T_3_ generation.

**Figure 17 ijms-24-05338-f017:**
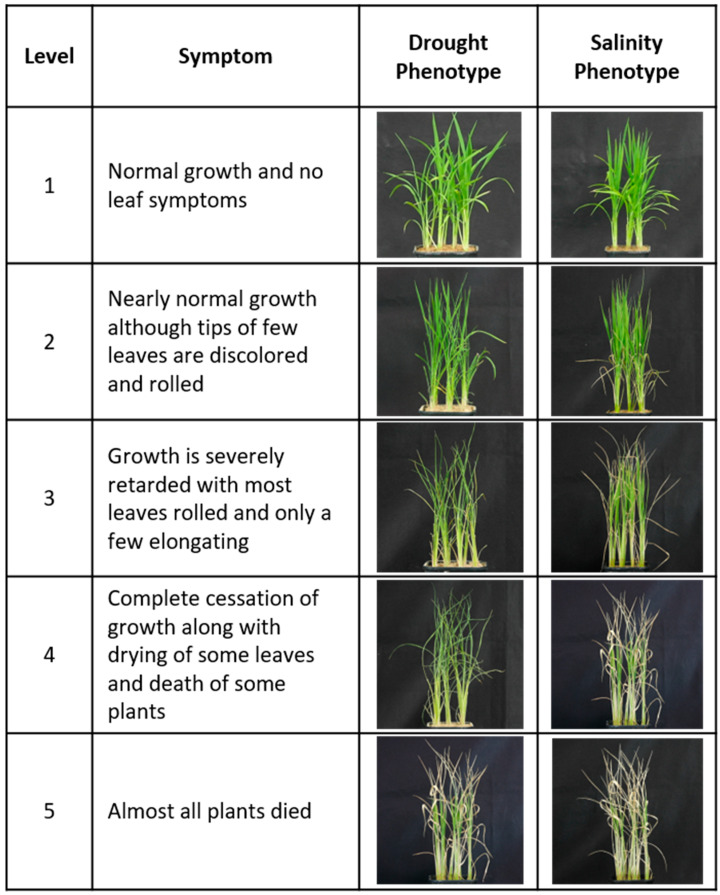
Description of scoring conditions for plant phenotypes after abiotic stress treatments. Description of the symptoms by each stress level.

**Table 1 ijms-24-05338-t001:** Analysis of *OsPUB7* transformation efficiency and mutation type ratio at the target site in T_0_ mutant rice plants.

Target Gene	No. of Plants Examined	No. of Plants with Mutations	Mutation Rate (%)	Putative Homozygous Mutations		Putative Bi-Allelic Mutations		Putative Heterozygous Mutations	
				No. of Plants	%	No. of Plants	%	No. of Plants	%
*OsPUB7*	31	15	48	4	26.7	6	40.0	5	33.3

**Table 2 ijms-24-05338-t002:** Chi-square (χ^2^) analysis of the genotypes with *bar* screening from *OsPUB7* T_1_ plants.

Gene	Line	Genotype	No. of Resistant Plants	No. of Susceptible Plants	No. of All Edited Plants	χ^2^	Significance (α)
*OsPUB7*	K343-7	Bi-allelic	267	54	321	11.449	0.003
	K343-8	Bi-allelic	130	42	172	0.031	0.07
	K343-17	Bi-allelic	23	16	39	5.342	0.001

**Table 3 ijms-24-05338-t003:** Agronomic traits of PUB7-GE in the T_3_ generation.

Gene	No. of Line	Plant Height (cm)	Culm Length (cm)	Panicle Length (cm)	No. of Tiller
Dongjin	WT	113.3 ± 1.3	94.5 ± 1.6	16.6 ± 0.9	12.4 ± 1.7
OsPUB7-GE	K343-7-5 ^ns^	115.1 ± 1.7 *	89.7 ± 0.6 *	21.6 ± 0.6 *	9.7 ± 0.5 ^ns^
	K343-8-1 ^ns^	111.9 ± 2.3 ^ns^	92.6 ± 2.7 ^ns^	19.0 ± 1.7 ^ns^	15.7 ± 1.0 ^ns^
	K343-17-8 *	118.5 ± 1.6 ^ns^	94.3 ± 1.2 ^ns^	18.5 ± 0.9 ^ns^	15.0 ± 0.8 ^ns^

*, *p* < 0.05; ns, not significant.

## Data Availability

Not applicable.

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
