# Peer review of "Knockout Mutants of OsPUB7 Generated Using CRISPR/Cas9 Revealed Abiotic Stress Tolerance in Rice"

_ijms, 2023, doi:10.3390/ijms24065338_

Round 1

Reviewer 1 Report

Authors submitted manuscript entitled “Knockout Mutants of OsPUB7 Generated using CRISPR/Cas9 Revealed Abiotic Stress Tolerance in Rice” to IJMS. In this work, authors used CRISPR/Cas9 technology to edit OsPUB7, and evaluated Cas9 mutants in response to abiotic stress. Finally, they obtained a stress-tolerant phenotype against drought and salinity stress and screened T-DNA-free lines. In addition, they found that mutants showed lower ion leakage and higher proline content than the WT. Moreover, by PPI network, they found the interaction of PUB7 with other positive and negative regulatory stress related genes. I have following comments and suggestions,

I’m unable to find Arabidopsis in fig 1.

L108; Where is the information about 25 genes

Revise the figure legend of fig 3 carefully. Provide the caption for each figure (e.g. a-c). The negative and positive control is missing in Gel picture. Why band size is different in Figure B?

Why authors have inserted green and red minus (-) symbols.

Please give name to each mutant in figure 4. e.g. pub7-1, pub7-2 etc. or same names as given in fig 5B.

In biallelic mutants the PAM site was also deleted? mutation was downwards. It is better if author can show the mutation types in chromatograms. Show chromatograms for all 4 homo, 6 bi-allelic mutants and 5 heterozygous mutants? What authors means here by bi-allelic and heterozygous mutants? What is the difference between bi-allelic and heterozygous mutants? Heterozygous mutants showed mutations in single allele?

Table 2. authors only tested biallelic mutant? Why?

Fig 8. Knockout mutants also showing normal PUB7 expression, even higher in different time points. Please make it clear. How the expression was significantly different at 0HAT in T & S genotypes? The histogram is not showing difference.

Have authors detected PUB7 expression in RNA-seq experiment? Authors should provide the accession number of RNA-seq files. Moreover, PUB7 and PUB8 show highly similar expression. I think these genes can have redundant functions.

The RNA-seq data is not properly described. No novel information is reported.

The discussion section is very weak. Authors have repeated statements from results section. Authors should speculate the results and discuss them as per their findings and already published studies. Moreover, provide statements about research gaps and possible future directions at the end.

Author Response

Reviewer comments 1

Authors submitted manuscript entitled “Knockout Mutants of OsPUB7 Generated using CRISPR/Cas9 Revealed Abiotic Stress Tolerance in Rice” to IJMS. In this work, authors used CRISPR/Cas9 technology to edit OsPUB7, and evaluated Cas9 mutants in response to abiotic stress. Finally, they obtained a stress-tolerant phenotype against drought and salinity stress and screened T-DNA-free lines. In addition, they found that mutants showed lower ion leakage and higher proline content than the WT. Moreover, by PPI network, they found the interaction of PUB7 with other positive and negative regulatory stress related genes. I have following comments and suggestions,

We appreciate the comments that the reviewers have given to our manuscript and the constructive criticism the reviewer has given. We have carefully reviewed the comments and have revised the manuscript accordingly. We believe that these changes have clearly improved our manuscript. 

I’m unable to find Arabidopsis in fig 1.

---Thank you for your critical comments. We revised Figure 1 to include Arabidopsis genes. Monocotyledonous crops such as rice and wheat and dicotyledonous crops such as Arabidopsis thaliana have structural differences, but it can be confirmed that their structures, such as the domain of PUB gene, are similar.

L108; Where is the information about 25 genes

---Thank you for your critical comments. We added Supplementary table 1 to supply information of 26 genes.

Revise the figure legend of fig 3 carefully. Provide the caption for each figure (e.g. a-c). The negative and positive control is missing in Gel picture. Why band size is different in Figure B?

---Thank you for your critical comments. We revised it in Figure 3 and Line 148 to 153 as follows: Figure 3. Development of transgenic rice plants with CRISPR/Cas9-OsPUB7 vector using Agrobacterium-mediated trans-formation. (A) Dongjin (WT) transformed with pPZP-3’Pinâ…¡-Bar vector containing CRISPR/Cas9::sgRNA gene. a, seeds plated on 2N6 media; b, callus formation; c, Agrobacterium co-culture in 2N6 AS media; d-e, multi-shoot differentiation; f, regenerated plants in rooting medium; g-h, acclimation in tissue culture room and planted in soil. (B) Detection of sgRNA/Cas9-mediated DNA modifications using PCR analysis of bar and nos terminator regions. L, DNA ladder; N, negative control (sterile water); P, positive control (plasmid vector).

Why authors have inserted green and red minus (-) symbols.

---Thank you for your critical comments. We revised it in Line 177 to 181 as follows: Figure 4. Identification of mutation genotypes in OsPUB7 generated using CRISPR/Cas9 technology using NGS analysis in Dongjin (WT) and T0 plants. Green indicated the target sequence, and blue indicated the PAM region. The InDel sequence of the target locus is shown in red. Deletion in the analysis region was indicated by "-", and insertion was indicated by an insertion of nucleotide such as "T" or "G". The minus (-) and plus (+) signs indicate the number of nucleotides deleted and inserted at the target sites, respectively. WT, wild type sequence with no mutations.

Please give name to each mutant in figure 4. e.g. pub7-1, pub7-2 etc. or same names as given in fig 5B.

---Thank you for your critical comments. We revised Figure 4 to supply information of mutants genotype.

In biallelic mutants the PAM site was also deleted? mutation was downwards. It is better if author can show the mutation types in chromatograms. Show chromatograms for all 4 homo, 6 bi-allelic mutants and 5 heterozygous mutants? What authors means here by bi-allelic and heterozygous mutants? What is the difference between bi-allelic and heterozygous mutants? Heterozygous mutants showed mutations in single allele?

---Thank you for your critical comments. We added all mutation types as shown Supplementary table 2 for 4 homozygous mutants, 6 bi-allelic mutants, and 5 heterozygous mutants. It was confirmed that PAM site was deleted in some mutants through NGS analysis. In eukaryotic cells, chromosomes are paired, so when a gene is knocked out, only one allele of a pair of genes is removed (mono-allelic knockout) or both alleles are removed (bi-allelic knockout). Bi-allelic in CRISPR knockout means that the mutation occurred in two chromosomes, but the two alleles are different.

Table 2. authors only tested biallelic mutant? Why?

---Thank you for your critical comments. Mutations that caused during double-strand breaking repair are predominantly deletions and 1bp insertion; these mutations are stably inherited in the progenies at both T2 and T3 generations following classic the Mendelian model. Once both copies of a target gene are mutated, as in a homozygous or bi-allelic edited plant, the genotype is stable and resistant to further editing by CRISPR/Cas9.  In previous report, “Mutations that occurred during double-strand breaking repair in plants appeared mainly as deletions and 1bp insertions. These mutations are stably inherited in offspring in both T2 and T3 generations according to the classical Mendelian model. When both copies of a target gene are mutated, as in homozygous or bi-allelic editing plants, the genotype is stable and resistant to further editing by CRISPR/Cas9. Bi-allelic gene targeting in seed-propagated plants has advantage of saving time in correcting genetic alterations (Feng et al., 2014; Endo et al., 2016).”

Fig 8. Knockout mutants also showing normal PUB7 expression, even higher in different time points. Please make it clear. How the expression was significantly different at 0HAT in T & S genotypes? The histogram is not showing difference.

---Thank you for your critical comments. We analyzed the relative gene expression level for significance based on 0HAT and revised Figure 8.

Have authors detected PUB7 expression in RNA-seq experiment? Authors should provide the accession number of RNA-seq files. Moreover, PUB7 and PUB8 show highly similar expression. I think these genes can have redundant functions.

---Thank you for your critical comments. We added Supplementary table 3, Supplementary table 4, and Supplementary figure 1 to provide information on RNA-sequencing analysis.

The RNA-seq data is not properly described. No novel information is reported.

---Thank you for your critical comments. We additionally added Supplementary figure 2 and revised in Line 327 to 336 to provide information on RNA-sequencing data that the RNA sequencing was carried out by ourselves with 12 samples.

The discussion section is very weak. Authors have repeated statements from results section. Authors should speculate the results and discuss them as per their findings and already published studies. Moreover, provide statements about research gaps and possible future directions at the end.

---Thank you for your critical comments. We have carefully revised 'Discussion' section in Lines 407 to 441.

Reviewer 2 Report

The article is multifaceted is extremely relevant and is of undoubted practical and scientific interest. It is undoubtedly a high-level work and the authors obtained significant very interesting results. The article is very interesting, original, the content of the article corresponds to the abstract and title. The tables and figures are complement the text well.

There are some comments and suggestions for authors.
1. The authors discusses the results of evaluating three lines K343-8-1, K343-17-8 and K343-7-5 for tolerance to drought and salt stress, but I advise explaining in more detail how they were selected from 15 lines carrying the mutation, why in the analysis took these lines?  Please explain this in the text. Please, add the section “plant material” to the materials and methods, give the scheme of the experiment in “Supplementary materials”, as, for example, it was done in the article - Feng et al., Multigeneration analysis reveals the inheritance, specificity, and patterns of CRISPR/Cas-inducedgene modifications in Arabidopsis Zhengyan. PNAS. 2014. 111(12). 632–4637 https://doi.org/10.1073/pnas.1400822111) (see figure 1 in this article).
2. It is also not clear why line K343-7-5 was not analyzed for tolerance to drought stress. In Figure 6 A. it is absent. It is advisable to add the evaluation data of the line K343-7-5 for tolerance to drought stress in Figure 6 A and in text, if it possible.

I recommend accepted the article for publication after making minor corrections.

Author Response

Reviewer comments 2

The article is multifaceted is extremely relevant and is of undoubted practical and scientific interest. It is undoubtedly a high-level work and the authors obtained significant very interesting results. The article is very interesting, original, the content of the article corresponds to the abstract and title. The tables and figures are complement the text well.

We appreciate the comments that the reviewers have given in our manuscript and the constructive criticism the reviewer has given. We have carefully reviewed the comments and have revised the manuscript accordingly. We believe that these changes have clearly improved our manuscript.

There are some comments and suggestions for authors.
1. The authors discussed the results of evaluating three lines K343-8-1, K343-17-8 and K343-7-5 for tolerance to drought and salt stress, but I advise explaining in more detail how they were selected from 15 lines carrying the mutation, why in the analysis took these lines?  Please explain this in the text. Please, add the section “plant material” to the materials and methods, give the scheme of the experiment in “Supplementary materials”, as, for example, it was done in the article - Feng et al., Multigeneration analysis reveals the inheritance, specificity, and patterns of CRISPR/Cas-induced gene modifications in Arabidopsis Zhengyan. PNAS. 2014. 111(12). 632–4637 https://doi.org/10.1073/pnas.1400822111) (see figure 1 in this article).

---Thank you for your critical comments. We added Supplementary figure 3 and Line 162 to 168 to provide information on the experimental process for selecting lines to undergo drought and salinity stress treatment.

  1. It is also not clear why line K343-7-5 was not analyzed for tolerance to drought stress. In Figure 6 A. it is absent. It is advisable to add the evaluation data of the line K343-7-5 for tolerance to drought stress in Figure 6 A and in text, if it possible.

---Thank you for your critical comments. We revised it in Figure 6 and Lines 207 to 208.  

I recommend accepted the article for publication after making minor corrections.

---Thank you for your critical comments. We have revised all minor points in the text of the manuscript.
